# Paraoxonases at the Heart of Neurological Disorders

**DOI:** 10.3390/ijms24086881

**Published:** 2023-04-07

**Authors:** Fatimah K. Khalaf, Jacob Connolly, Bella Khatib-Shahidi, Abdulsahib Albehadili, Iman Tassavvor, Meghana Ranabothu, Noha Eid, Prabhatchandra Dube, Samer J. Khouri, Deepak Malhotra, Steven T. Haller, David J. Kennedy

**Affiliations:** 1Department of Medicine, University of Toledo College of Medicine and Life Sciences, Toledo, OH 43606, USA; 2Department of Medicine, University of Alkafeel College of Medicine, Najaf 54001, Iraq; 3Department of Computer Engineering Technology, College of Information Technology, Imam Ja’afar Al-Sadiq University, Najaf 54001, Iraq

**Keywords:** paraoxonases, neurodegenerative diseases, Parkinson’s disease, Alzheimer’s disease

## Abstract

Paraoxonase enzymes serve as an important physiological redox system that participates in the protection against cellular injury caused by oxidative stress. The PON enzymes family consists of three members (PON-1, PON-2, and PON-3) that share a similar structure and location as a cluster on human chromosome 7. These enzymes exhibit anti-inflammatory and antioxidant properties with well-described roles in preventing cardiovascular disease. Perturbations in PON enzyme levels and their activity have also been linked with the development and progression of many neurological disorders and neurodegenerative diseases. The current review summarizes the available evidence on the role of PONs in these diseases and their ability to modify risk factors for neurological disorders. We present the current findings on the role of PONs in Alzheimer’s disease, Parkinson’s disease, and other neurodegenerative and neurological diseases.

## 1. Introduction to the Paraoxonase Enzyme Family and Brain Tissue Distribution

In humans, the Paraoxonase (PON) family of enzymes (PON-1, PON-2, and PON-3) are encoded by three adjacent genes located on human chromosome 7q21.3. These three enzymes share similar activities and have a 90% structural similarity. The PONs are calcium-dependent esterases with an approximate molecular mass of 40–45 kDa; additional structural properties are reviewed in [1]. PON-1 and PON-3 are mainly synthesized in the liver and circulate bound to high-density lipoprotein (HDL) in serum, whereas PON-2 is synthesized locally in tissues such as the brain, kidney, liver, and testis [2,3,4]. We and others have demonstrated well-established roles for diminished expression and activity of PONs in cardiovascular and renal disease both clinically and experimentally, and this has been previously reviewed [1]. Clinically, PON-1 has well-known cardioprotective roles in patients with stable coronary artery disease [5], systolic heart failure [6], stable chronic heart failure [7], and chronic kidney disease [6,8]. Experimentally, PON-1 not only has cardioprotective roles in chronic kidney disease [9], but also renal anti-inflammatory and anti-fibrotic roles in the setting of chronic hypertension [10] and renal ischemia [11]. Additionally, PON-2 has demonstrated a cardioprotective role, which may be associated with its ability to improve mitochondrial function and diminish reactive oxygen species generation [12]. Furthermore, PON-3 has recently been shown to participate in the metabolism of cardiotonic steroids in settings such as hypertension and chronic kidney disease [13]. In the current work, we review the important roles PONs play outside of these established cardiovascular functions by examining a variety of roles for PONs in common neurodegenerative diseases as well as other associated neurological disorders.

Tissue expression of PON-1 and PON-3 has been reported in minimal quantities that vary in different tissues. Using a bioinformatic approach to analyze publicly available tissue-specific RNA-seq data from both human and murine brain, we noted that PON-2 is expressed at higher levels in human brain tissue than PON-1 and PON-3, with astrocytes showing the highest expression levels of the PON enzymes, followed by oligodendrocytes (Figure 1A). We observed similar results in murine brain tissue, except that microglia record the highest expression levels of PON-3 (Figure 1B). Our analysis agrees with Giordano et al., who indicated, by Western blot, that PON-2 is expressed at significantly higher levels than PON-1 in all brain areas. PON-3, on the other hand, was not detected in any areas of the murine brain [14,15]. PON enzymes can metabolize lactone compounds, glucuronide drugs, nerve gases, aromatic carboxylic acid, aryl esters, and some carbamate insecticides. Additionally, they inhibit the lipoxidation of low-density lipoprotein (LDL) metabolites [16]. A study by Salazar and coworkers showed that PON-1 and PON-3 can cross the blood–brain barrier and transfer to specific brain-cell types. This study showed the localization of PON-1 and PON-3 around and inside Aβ plaques in a murine model of Alzheimer’s disease (AD), suggesting that HDL particles carry PON-1 and PON-3 from the liver, where they are synthesized to areas of high levels of inflammation and oxidative stress. This signifies the PON enzymes’ potential role in attenuating oxidative stress and lipid peroxidation in AD and other neurodegenerative diseases (ND) [17]. Additional studies have highlighted the putative role of PON enzymes in brain health and disease [15,18,19], which demonstrated a synergistic effect against ND progression.

## 2. Overview of PON-1 and Its Neurological Associations

Neurodegenerative diseases (ND) are a broad range of disorders characterized by neuronal tissue damage and loss of function, leading to cell death [20]. The etiology of ND is multifactorial and still not fully understood [21]. However, many of these disorders seem to have common cellular and molecular mechanisms underlying the pathogenesis; for example, certain toxin exposures, increased oxidative stress, and decreased antioxidant activity [22]. PON enzymes serve as an important physiological redox system which participates in protecting against cellular injury caused by genotoxic and oxidative damage. PON-1 is a hydrolytic lactonase enzyme that is synthesized in the liver and circulates bound to HDL. It provides the antioxidant property that prevents LDL and HDL oxidation and contributes to much of the anti-oxidative and anti-atherogenic activities that have been attributed to HDL [23,24]. It also protects HDL and LDL from oxidative stress through the elimination of ROS produced by metabolism. PON possesses peroxidase-like activity that can contribute significantly to the protective effect of PON against lipoprotein oxidation [25]. PON enzyme activity in serum has been correlated with protection against oxidative damage [26,27]. Oxidative stress plays an essential role in the pathogenesis of many ND, even though the exact mechanistic links are not fully elucidated. Nevertheless, many studies have highlighted the links between PON-1 and ND progression [28,29,30].

PON-1 protects against atherogenesis by metabolizing oxidized lipids. Its essential role as a protective factor against atherogenesis continues to attract more attention in epidemiological studies [31,32]. Studies have demonstrated PON-1’s role in ischemic stroke, one of the top neurological disorders linked with atherosclerosis [33,34,35,36,37]. Beyond stroke, pesticide exposure has consistently been associated with the development of ND [38,39]. Exposure to pesticides results in induced oxidative stress, mitochondrial dysfunction, and impairment of the ubiquitin–proteasome system, mechanisms that are related to neuronal cell death in ND [40]. PON-1 detoxifies xenobiotics, including pesticides, and also hydrolyses bioactive toxic oxon metabolites, such as parathion, diazinon, and chlorpyrifos, converting them into nontoxic metabolites [41,42]. In humans, serum PON-1 levels and activity show up to a 40-fold interindividual variation and can be genetically influenced by common polymorphisms of the PON-1 gene [43]. Several polymorphisms have been identified in the promoter and coding regions for the PON-1 gene. The coding region polymorphisms include a methionine/leucine substitution at position 55 (Leu-Met-55) and an arginine/glutamine substitution at position 192 (Gln-Arg 192) [44]. Both have been shown to influence serum levels and biological activity. Studies report the PON-1 L55M polymorphism as a risk factor in AD while the Q192R polymorphism demonstrated a protective function [45]. On the other hand, studies have also found that the presence of the PON-1 R192 allele raises the risk of cardiovascular disease [46]. Further, it was observed that the PON-1 L55M polymorphism causes a decrease in PON-1 levels while the PON-1 Q192R mutation causes an elevation in enzyme levels. Therefore, PON-1 may serve as a potential biomarker for determining the severity and prognosis of ND in subjects with different genotypes. Additional clinical studies indicate that low PON-1 activity could be a potential risk factor for ND. Because PON-1’s esterase, lactonase, and arylesterase activities are significantly affected by its polymorphisms, considerable attention has been devoted to understanding the role of PON-1 in the emerging risk of ND. Here, we examine the molecular mechanisms of PON-1 status as it relates to these disorders.

### 2.1. PON-1 and Parkinson’s Disease

Parkinson’s disease (PD) is a progressive ND involving motor and neural abnormalities. This disease stems from the degeneration of a nucleus in the human brain located within the mesencephalon known as the substantia nigra. Specifically, PD involves the disintegration of the dopamine-producing neurons within the substantia nigra. Several polymorphisms of PON-1 have been discovered and studied as having effects that may promote the development of PD. As mentioned previously, PON-1 has an essential role in detoxifying organophosphate (OP) compounds, a class of chemicals that includes insecticides. Hence, PON-1 has an essential role in the prevention of some disorders like PD (Figure 2). Histidine and cysteine residues in the active site of PON-1 are important for its arylesterase activity. Studies show that the rates of substrate hydrolysis are significantly different between the two PON-1 polymorphisms. The PON-1 R192 substitution appears to affect the substrate-dependent activity seen among individuals with different PON-1 polymorphisms. For instance, PON-1 R192 hydrolyzes paraoxon more efficiently than PON-1 Q192R, whereas PON-1 Q192R hydrolyzes diazoxon, sarin, and soma faster than PON-1 R192. On the other hand, the L55M substitution appears to show no effect on the rates of different substrates’ hydrolysis. PON-1 M55 was reported to be associated with lower protein level, activity, and mRNA expression. Hence, people with certain PON-1 polymorphisms might be susceptible to organophosphate toxicity. Indeed, organophosphate-based insecticide exposure has known harmful effects on agricultural workers, including DNA damage, which is elevated seasonally during periods of insecticide application. PON-1 serves as the main means of protecting the nervous system against the neurotoxicity of organophosphates [47]. Exposure to organophosphates, such as pesticides and other environmental toxins, is linked to the development of neurological disorders in which acetylcholine has a significant role, including PD. An analysis by Mota et al. indicated that those who work with organophosphates are at a significantly increased risk for PD. They found that the PON activity, but not its arylesterase activity, may be causally involved in the progression of PD [47]. As mentioned above, PD involves the degeneration of the substantia nigra, which further generates an imbalance of dopamine and acetylcholine. Because of this, free radical species generate oxons that attack and inhibit acetylcholinesterase, which increases the amount of acetylcholine, exacerbating this imbalance. However, the peroxidase and triesterase actions of PON-1 can attenuate these effects and prevent the worsening imbalance in PD [48] (Figure 2).

It is now apparent that genetic variation in the PON-1 enzyme may modify the risk of PD [44]. In a study conducted by Lee et al., several functional PON-1 variants were suggested to modify pesticide-induced PD risk, including L55M and Q192R. Carmine et al. performed a study particularly focused on the association of the PON-1 Met-55 allele with PD. The basis of this study rested on the fact that the L55M polymorphism in PON-1 is exclusively associated with both mRNA and protein levels, where the Leu allele is known to produce more PON-1 mRNA than the Met allele. Consequently, carriers of the Met-55 allele may have an inherited defect in the detoxification of environmental toxins and may also have an increased susceptibility to PD [44]. In a Swedish study, exon 3 was sequenced in 114 PD patients and 127 control subjects to determine the distribution of the L55M polymorphism in PON-1 [44]. The results confirmed that there is an association between the L55M polymorphism and PD. A similar Swedish study by Belin et al. agreed that the L55M polymorphism in the PON-1 gene is associated with PD. The findings of this study confirmed that higher PON-1 levels reduce the risk for PD [49]. Additionally, this study found that the minor rs854751 PON-1 promoter polymorphism allele suggests a more protective effect of the enzyme as it was more common among healthy control subjects. Belin et al. further suggests that this minor allele can increase levels of PON-1 expression and decrease the risk for PD. An alternative study conducted by Akhmedova et al. established that the Met-55 allele of PON-1 was significantly higher in PD patients than in control subjects [50]. They further concluded that this allele is possibly an independent risk factor for PD. Therefore, a mutation of this allele could trigger a PON-1-lessened metabolism of environmental neurotoxins and may have a role in neurodegeneration. Additionally, a Japanese study done by Kondo and Yamamoto looked into a biallelic PON-1 polymorphism at codon 192 (A and B alleles) in 166 patients with PD [51]. This study concluded that the frequency of the B (Arg) allele in PON-1 was significantly increased in PD patients compared to controls. They also discovered that the risk of PD in homozygotes for the B allele was 1.6 times higher than those with the A (Gln) allele [51]. This study further determined that homozygotes for the B allele are poor metabolizers of environmental toxins that may contribute to the neurodegeneration seen in PD. Nevertheless, the relationship between PON-1 and PD is complex and multi-factorial and it is clear that additional clinical and experimental research is needed to decipher whether the aforementioned genetic differences represent gain or loss of PON function and whether these associations are causally linked to PD.

### 2.2. PON-1 and Alzheimer’s Disease

Alzheimer’s disease (AD) is the most common form of dementia and is an irreversible and progressive ND that impairs neurological functions such as memory and cognition. As the disease progresses, patients with AD lose the ability to perform even simple tasks. The exact cause of AD has yet to be fully understood, though research has shown that combinations of multiple factors, such as genetics, lifestyle, and environment, possibly contribute to the pathological changes in the brain. In AD, healthy communication between neurons is disrupted, leading to loss of function and cell death. It is hypothesized that in AD, damaged neurons, neurites, highly insoluble Aβ deposits, and neurofibrillary tangles stimulate inflammation and perhaps even an autoimmune response [52,53]. Aβ may also promote the formation of atherosclerotic damages through oxidative and pro-inflammatory processes within vessels [54]. Hypoperfusion and hypoxia due to blocked atherosclerotic cerebral vessels worsen Aβ polymers, thus worsening the clinical outcome [55]. This review highlights the connection of PON enzymes and AD and explores the role of PON-1 in the progression of AD. As previously described, PON-1 is an esterase associated with apolipoprotein AI and clusterin carried by HDL In a study of autopsy-confirmed AD cases, the L55M and Q192R ploymorphisms were associated with beta-amyloid levels [56]. Because of PON-1’s esterase, lactonase, and arylesterase activities, previous PON studies have considered the enzyme Das as a first-line and natural protective agent against organophosphates, conveying atheroprotective and inflammatory effects [57]. However, Erlich et al. indicate that PON-1 is involved in a mechanistic pathway that preserves brain integrity. Their study reports PON-1 enzymatic activity as a key association in AD patients with different phenotypes of AD compared to non-demented controls in three different ethnic groups. Hence, low PON-1 activity could be a potential risk factor for ND [58]. These results show that PON-1 activity was reduced in patients with AD or mixed dementia when compared with control subjects. Furthermore, homocysteine levels were elevated in both types of dementia. The negative association between PON-1 activity and homocysteine levels points to a significant role for PON-1 in the pathogenesis of AD. Comparable findings were observed in a study that involved patients with either AD or vascular dementia. PON-1 activity was significantly lower in both patient groups compared to the healthy control subjects. A mice model study showed that PON-1 deficiency and elevated homocysteine levels change the crucial balance of protein expression involved in neural degeneration and plasticity and other important neural functions, such as development, learning, and aging [59]. The findings suggest a novel role for PON-1 as a modulator of the brain proteasome and explain a possible mechanism for PON-1 in ND (Figure 3). Several human clinical studies have shown that low serum PON-1 activity leads to an increased risk for dementia and AD [60,61,62]. Paragh et al. illustrate a link between cholesterol level and the progression of AD and vascular dementia (VAD) with a change in PON-HDL enzyme activity in the AD group compared to normal subjects. Serum cholesterol was significantly higher in patients with AD and VAD compared to controls. PON-1 HDL-associated activity in AD and VAD was significantly lower than in healthy controls, and the PON/HDL ratio in those patients was decreased compared to healthy controls [63]. In their clinical study, Cervellati et al. also indicate that mild cognitive impairment (MCI), VAD, and late-onset AD patients had lower arylesterase and PON-1 levels than healthy controls. Additionally, Saeidi et al. compared PON-1 arylesterase activity, genotype, and allele frequency in healthy subjects with late-onset AD subjects and found that PON-1 activity is significantly lower in AD subjects when compared to controls [56]. PON-1 mechanistic enzymatic activity needs to be further explored through basic science and human clinical studies; however, its activity can be inferred to be the main means of protecting the central nervous system against AD and other forms of dementia. Recent studies have shown PON enzymes, specifically PON-1, play an essential role as anti-inflammatory and anti-oxidative stress modulators against foam cell formation in cardiovascular diseases. Chronic inflammation activates macrophages, which play an important role in atherosclerotic lesion formation, where they actively participate in cholesterol accumulation. Concerning AD, as the disease progresses, inflammation and oxidative damage play a major role in worsening patient outcomes. However, it is unclear if and how PON-1 could help through its anti-inflammatory and antioxidant properties to protect against later stages of AD. Mice model studies in recent years have shown for the first time that PON-1 can have anti-inflammatory protective properties. Aharoni et al. indicate that PON-1 provides feedback inhibition on pro-inflammatory molecule production, demonstrating PON-1 binding to innate inflammatory molecules and increasing resistance against inflammatory molecules such as TNFα and IL-6. This further illustrates the protective impact of PON-1 on innate immune cells such as macrophages that classically need pro-inflammatory activation [57]. Therefore, reducing macrophage accumulation in atherosclerotic lesions could prevent worsening and the blocking of atherosclerotic cerebral vessels. Thus, potentially preventing worsening Aβ polymers and the formation of more atherosclerotic damage through oxidative and pro-inflammatory processes within vessels may play a role in preventing impaired clinical outcomes (Figure 3). Consistent with this thought, Narasimhulu et al. illustrate that ApoE-PON-1-deficient mice have AD’s biochemical and morphological characteristics. Aged ApoE-PON-1-deficient mice were shown to have consistent AD-specific biomarkers such as elevated S100B protein and presenilin 1 and 2, which other studies have shown to be caused by neuroinflammation and loss of blood–brain barrier integrity [59]. In a clinical study, Bachetti et al. reported that systemic oxidative stress in AD patients shown to have lower PON-1 activity could contribute to inflammation and oxidative damage [60]. From this experimental and clinical data, it appears that PON-1 may specifically play a major role in the protection against the pathophysiology and progression of AD. Due to the complex nature of AD and PON-1, whether PON-1 plays a direct or indirect role in regulating the neuroinflammation and oxidative stress which accompany AD initiation and progression warrants further study.

### 2.3. PON-1 and Neuroinflammation

The topic of neuroinflammation can be somewhat challenging to define, but on the most superficial level, neuroinflammation is an inflammatory response within the brain or spinal cord driven by the production of cytokines, chemokines, and reactive oxygen species (ROS) [64]. While most understand neuroinflammation as a negative response leading to cell death and tissue damage, low amounts of neuroinflammation are crucial for brain function and tissue repair. However, when this neuroinflammatory response is prolonged, we often see many of the adverse effects of neuroinflammation, such as neuronal damage, cognitive impairment, and reduced plasticity [64]. For the purposes of this review, we will focus on the negative aspects of neuroinflammation and how PON-1 may regulate it. DiSabato et al. demonstrate that microglia are the key players in neuroinflammation. These immune cells perform macrophage-like activities for the central nervous system [64]. In a diseased state, microglia become activated and begin to produce inflammatory chemokines and cytokines. If this process is chronic, the result is often the negative aspects of neuroinflammation discussed above. We know that reduction in PON-1 level and/or activity is linked to oxidative stress, inflammation, and atherosclerosis, but recent research has shown that it may also be related to mild cognitive impairment [28]. Menini and Gugliucci show that PON-1′s antioxidant and anti-inflammatory properties may serve a protective role in neurovascular disease [48]. PON-1 appears to provide a counter-regulatory response to ROS in a near ubiquitous fashion. Levy et al. show that the presence of PON confers a protective effect against neuroinflammation due to its ability to eliminate ROS [65]. The reduction of PON-1 may lead to the chronic effects seen with prolonged microglia activation in neuroinflammation [64]. Overall, through its anti-inflammatory properties, PON-1 and the other PON enzymes may serve a protective role against prolonged neuroinflammation that leads to adverse outcomes, such as cognitive impairment, reduced plasticity, and neuronal damage.

### 2.4. PON-1 and Motor Neuron Diseases

Amyotrophic lateral sclerosis (ALS) is an ND that causes progressive muscle weakness and ultimately leads to death by respiratory failure. Also known as motor neuron disease (MND), the fatal condition results from the loss of upper and lower motor neurons that innervate skeletal muscles [66]. The disease affects about 2 per every 100,000 people each year, with a typical onset between the ages of 35 and 50 and a life expectancy of 3–5 years [67]. Current evidence indicates that ALS is heritable, as recent studies have shown that various genetic defects may contribute to familial ALS. The genetics of the more common sporadic condition are not very well understood, as no sole gene has been confirmed to increase the risk of ALS. Several risk factors lead to sporadic ALS, with the most common being exposure to toxic environmental factors. The best known of these toxins are organophosphates, found in chemical warfare agents and insecticides/herbicides [68]. These chemicals can be metabolized through oxidative pathways. Because the exact pathogenesis of this disease remains elusive, it has been challenging to find a definitive treatment to prevent the process of neurodegeneration. A significant enzyme thought to play a role in the development of ALS is PON-1. This enzyme functions in metabolizing the organophosphates that potentiate neurotoxic activities in the voluntary motor system. Likewise, PON-1 protects against oxidative stress in the lipids of neuronal membranes. Due to its protective mechanisms against common pathways involved in the pathogenesis of ALS, it is believed that mutations in PON-1 can increase susceptibility to environmental toxins. In fact, numerous studies investigating the relationship between PON-1 activity and sporadic ALS suggest that defects in the PON-1 gene play a role in ALS development [69,70]. However, a meta-analysis from multiple large-scale international studies indicate that there is no clear correlation between PON-1 and ALS [71].

The potential pathogenic link between PON-1 expression/activity and the development of ALS is not well understood. Some studies suggest a correlation between PON-1 defects, such as polymorphisms, and increased vulnerability to ALS. Additionally, other research shows that defects in the gene encoding PON-1 may increase the risk of developing sporadic ALS. One study primarily focused on the frequency of the PON-1 variants 192Q and 192R allele in patients with ALS. The prevalence of this single-nucleotide polymorphism was assessed in 409 ALS patients [72]. Patients were screened for their genotype, and the SNP Q192R was analyzed. Using a dominant model, they found that the minor allele G was linked to bulbar onset (30.9% vs. 24.6%) and independent for reduced survival (OR = 1.38). These results indicate that PON-1 may play a role as a disease modifier gene in sporadic ALS. The SNP was associated with a reduced detoxifying function of PON-1, promoting the ALS phenotype. Similarly, another study proposed that the downregulation of PON-1 reduces xenobiotic detoxification systems, which leads to adult-onset ND [69]. Individuals with PON-1 defects are believed to be at a greater risk of environmental toxin poisoning, which is a hallmark of sporadic ALS. This study also found that PON-1 was reduced in the peripheral tissues of ALS patients. Additionally, the gene expression of PON-2 was analyzed in ALS patients, and it was found that this enzyme was reduced in patients with ALS compared to non-affected, healthy subjects. Another study explored 20 SNPs within the PON gene cluster and assessed their significance to ALS [73]. The study examined 597 individuals with PON SNPs and 692 control subjects for the relation to ALS and ALS sub-phenotypes. Results indicated that two common SNPs, rs987539 and rs2074351, are potentially involved in an increased vulnerability to ALS. However, it was revealed that none of the 20 SNPs have a clear correlation with age-onset or survival. The study further suggested an association between sporadic ALS and promoter haplotypes that reduce PON-1 levels. It was also found that control subjects share a link with the haplotypes that increase gene expression. However, there was no significant association between coding polymorphisms at the haplotype level and ALS This study also focused on the interactions between genes and the environment and concluded that some PON-1 promoter mutations might increase the susceptibility to developing ALS from exposure to organophosphate toxins. A similar study investigated the efficacy of PON-1 deviants in detoxifying organophosphate toxins [74]. These gene–environment interactions were cross-analyzed with 143 control subjects and 143 sporadic ALS patients to evaluate the potential relationship between the two. Polymorphisms L55M, Q192R, and I102V found in PON-1 coding regions and polymorphisms -909c>g, -832g>a, -162g>a, and -108c>t found in the promoter region were genotyped in each individual. They determined that reduced PON-1 expression, resulting from promoter allele _108t, significantly related to sporadic ALS. Furthermore, two additional studies focused on the development of sporadic motor neuron diseases (MNDs) due to exposure to neurotoxic substances like organophosphates. These studies prioritize organophosphate-induced delayed neuropathy (OPIDN), an ND similar to ALS, in that both conditions are characterized by ataxia and progressive paralysis. OPIDN is suggested to be initiated through the inhibition and subsequent aging of neuropathy target esterase (NTE). NTE is a lysophospholipase/phospholipase B that regulates phospholipid equilibrium. NTE is found in the nervous system and is inhibited by organophosphorylation. Mutations in NTE are found to lead to spastic paraplegia in MND. This study reflects how PON polymorphisms influence vulnerability to ALS due to gene–environment interactions [75,76]. However, multiple reviews propose that there is no absolute association between PON-1 and ALS. They criticize the number of confounders not considered in previous experiments, including the lack of evidence in reducing serum PON-1 activity, regardless of high-frequency polymorphisms [66]. Even though SNPs are presumed to reduce the enzyme activity of metabolizing organophosphates, this study shows that these effects depend on the substrate. This underscores the complexity of PON-1 functionality, where enzyme activity and serum concentration may vary based on lifestyle and demographic factors such as smoking and age. The variability with this enzyme makes it difficult to establish a clear association between PON-1 activity and sporadic ALS [66]. A meta-analysis of the impact of PON-1 Q192R and L55M polymorphisms on ALS was carried out to explore further the relationship between the two variables across numerous studies. Seven studies of the PON-1 L55M polymorphism and eight studies of the PON-1 Q192R polymorphism were analyzed, totaling 2831 ALS cases and 3123 control subjects [77]. The meta-analysis found no clear association between the PON-1 variants (PON-1 55M and 192R) and the development of ALS. The analysis considered recessive, dominant, and homozygous contrast models and stratification by ethnicity, neither of which pointed to a close link between defects in the PON-1 gene and ALS development [77]. In two other studies, researchers conducted a meta-analysis on ten published studies and one unpublished study to investigate the relationship between PON-1 and PON-3 genetic variants and sporadic ALS [78,79]. In a case–control study that included 1160 patients with sporadic ALS and 1240 control samples, the PON-1 and PON-3 SNPs were analyzed for their frequencies in each subject. Results showed that the frequencies of SNPs are not significantly different across the test and control groups. PON mutations were observed in 2.1% of familial ALS patients and 1.4% of sporadic ALS patients. There was little difference in control subjects, as 2.5% of mutations were identified in the control group. The studies concluded that PON-1 SNPs are not directly linked to ALS, and if they do factor into the disease, it most likely affects patients that are most vulnerable to its development. Similarly, another study carried out a separate meta-analysis on ten published studies and one unpublished study of the PON mutations to assess their link to ALS. The study investigated this relationship in 4037 ALS patients and 4609 control samples. Additionally, genome-wide association data were evaluated in 2018 ALS patients compared to 2425 control subjects and found no significant association between ALS development and polymorphisms in the PON locus [80]. Therefore, further work is needed to elucidate the true role of PON-1’s relation to the development and pathogenesis of ALS.

### 2.5. PON-1 and Brain Tumors

Little research exists on the relationship between PON-1 and brain tumors. However, a few studies have investigated the causative association between PON-1, astrocytomas, and meningiomas. One study assessed the risk of developing astrocytomas and meningiomas because of PON-1 polymorphisms. A total of 71 cases, 43 with astrocytoma and 28 with meningioma, were compared to 220 healthy control samples for frequency of the PON-1 variants L55M and Q192R. Results showed that the frequency of mutations between the patients and the control subjects did not differ and the common nonsynonymous PON-1 polymorphisms did not share a clear association with brain tumor development [81]. A second study also evaluated the oxidative response from PON-1 polymorphisms and its effect on adult brain tumor development. Data were obtained from the National Cancer Institute from non-Hispanic white populations. The risk of glioma (*n* = 362), meningioma (*n* = 134), and acoustic neuroma (*n* = 69) were compared to healthy controls (*n* = 494). The SNP frequency from a number of key anti-oxidant genes, including CAT, GPX1, NOS3, PON-1, SOD1, SOD2, and SOD3, were analyzed. Overall, the results showed that PON-1 serum levels were lower in glioma and meningioma cases than in controls. However, no clear relationship between brain tumors and the Q192R polymorphism was confirmed. It was ultimately suggested that common variants in the genes SOD2, SOD3, and CAT might have a greater influence on brain tumor risk [82]. A similar study evaluated the PON-1 Q192R polymorphism expression in high-grade gliomas and meningiomas [83]. Forty-two high-grade gliomas and forty-two meningiomas were compared to fifty non-cancer control subjects for their serum PON-1 activity. In this study, PON-1 expression was reduced in brain tumor patients compared to non-cancer patients (*p* < 0.001), but PON-1 serum levels were similar across gliomas and meningiomas. Overall, these results show that PON-1 polymorphisms may be associated with tumorigenesis of the brain, but additional study is needed.

## 3. Overview of PON-2 and Its Neurological Associations

Paraoxonase 2 (PON-2) is the oldest member of the PON family. Compared to the other two PON genes, PON-2 is found in many tissues throughout the body, with high expression in the brain, heart, and lungs [84]. PON-2 uses calcium to hydrolyze lactones, esters, and aryl esters and functions as an antioxidant, reducing the levels of ROS. In addition, PON-2 is found in the endoplasmic reticulum (ER) and mitochondria, binding to coenzyme Q10 and preventing superoxide formation [85].

PON-2 plays an important role in the brain and has critical neuroprotective properties. Its expression has been found to be the highest in dopaminergic regions, specifically the nucleus accumbens, striatum, and substantia nigra. Astrocytes, in comparison to neurons, have significantly higher levels of PON-2, and a deficit of expression in both cell types can lead to high levels of oxidative stress and the inability to recover from toxicity, which can subsequently lead to death [15].

Sex and age can be major determinants of PON-2 expression. Since PON-2 has a critical role in the regulation of oxidative stress and is an anti-inflammatory factor, it is important to consider sex and age as variables when assessing PON-2 expression [15,86]. The association between polymorphisms of PON-2 and enzymatic activities in the neurodegeneration process still needs to be understood. Some studies show that low levels of PON-2 expression, due to its potent neuroprotective characteristics, impact various neurodegenerative diseases and conditions, such as AD, PD, ALS, and cerebral ischemia-reperfusion injuries [69,87,88,89]. Moreover, PON-2 may be a potential therapeutic target in brain tumors, as it can aid in modulating the levels of oxidative stress, apoptosis, and cellular proliferation in tumors [90].

### 3.1. PON-2 in Neurodevelopment

During neurodevelopment, it is important that the brain is protected from oxidative stress, which can damage cells and significantly impair their function. Giordano et al. found that PON-2 levels and expression in the brain slowly increase after birth and peak in the neonatal stage of life in murine models [15,86]. They also noticed that the expression levels gradually decrease with aging, signifying that the high levels of PON-2 earlier in life can aid in protecting the neuronal tissue in the brain from oxidative stress, especially during the developmental stages. In addition, Garrick et al. suggest that decreased levels of PON-2 in the brain early in development may lead to neurological damage by oxidants in later developmental stages, increasing susceptibility to diseases like AD and PD [86].

### 3.2. PON-2 in Alzheimer’s Disease

AD is characterized by an increase in ROS and cholinergic deficiency, resulting in acetylcholinesterase inhibitors (AChEIs) as a treatment option for those with the condition. Parween et al. (2021) found that PON-2 is responsible for the hydrolysis of donepezil hydrochloride (DHC) and pyridostigmine bromide (PB), two AChEIs. Additionally, PON-2 polymorphic mutants display increased esterase activity, resulting in the ineffectiveness of these drugs and increasing the variability of the responses to therapy, which could be dependent on the levels of PON expression [91,92]. Moreover, Khan et al. researched how quercetin, a plant flavonoid, has antioxidant properties including induction of PON-2, which can be beneficial to administer to those with AD as it enhances the body’s neuroprotective mechanisms [88]. The researchers found that the neuroprotective effect of quercetin was significantly diminished in PON-2 knockout mice when compared to controls, suggesting the importance of PON-2 for quercetin’s function. There are several hypotheses for how quercetin interacts with PON-2, but the exact mechanism is unknown. One potential mechanism is that quercetin induces PON-2 expression through the JNK/AP-1 pathway and the other is that, due to quercetin phytoestrogen activity, PON-2 expression is increased [91,93,94]. Additional study will allow the discovery of the true role of PON-2 in the neuroprotective role of quercetin.

### 3.3. PON-2 in Parkinson’s Disease

The development of PD is characterized by increased levels of dopamine and loss of function mutations in the DJ-1 (PARK7) gene. DJ-1 mutations account for 1% of familial PD, which means that such mutations are very rare [89]. Parsanejad et al. examine how DJ-1 interacts with PON-2 to provide neuroprotective properties in the brain. The researchers found that DJ-1 knockout cells have less PON-2 activity and respond less to oxidative stress when compared to the controls, suggesting that DJ-1 is a critical factor in regulating PON-2 activity [89]. In addition, since PON-2 expression is the highest during neonatal development and decreases during adulthood, this makes neuronal cells more susceptible to oxidative stress and toxins, like parkinsonian toxin 1-methyl-4-phenyl-1,2,3,6-tetrahydropyridine (MPTP). Moreover, lack of PON-2 can cause neurons to face a great amount of oxidative stress, which can be induced by 1-methyl-4-phenylpyridinium (MPP+), a toxin produced after MPTP delivery. Therefore, it is important to consider the role of PON-2 in PD, as it has been proposed as a potential therapeutic target [95].

### 3.4. Hormonal Regulation of PON-2 in Neurodegenerative Disorders

Understanding the hormonal regulation of PON-2 can further explain the gender differences seen in neurodegenerative disorders. PON-2 expression appears to be positively modulated by estrogens; therefore, it is higher in CNS tissues of females than males when examined in mice. In every brain region, PON-2 levels are higher in female mice than in male mice, which was confirmed by measurements of lactonase activity, using dihydrocoumarin (DHC) hydrolysis, and of PON-2 mRNA levels [86]. Cheng and Klaassen examined female mice that were ovariectomized and found that PON-2 mRNA and protein levels decreased when compared to that of normal male and female mice [96]. Because females normally have more estrogen, males are more likely to be affected by oxidative stress and neurodegenerative diseases such as PD. This could explain why the incidence of PD is 90% higher in males when compared to females, highlighting that lower PON-2 levels in dopaminergic neurons of males may not provide the necessary protection needed against oxidative stress [91].

### 3.5. PON-2 in Cerebral Ischemia-Reperfusion Injury

Ischemia in the brain can be due to a sudden occlusion of an artery through thrombosis or an embolism. To salvage ischemic tissue, reestablishment of the blood flow is important; however, rapid blood flow can lead to injury in the brain and cause cerebral ischemia-reperfusion injury [87]. Regarding its pathogenesis, one of the main causes is elevated ROS and subsequent oxidative stress. One study investigated the potential neuroprotective function of PON-2 in cerebral ischemia-reperfusion injury, using oxidative stress caused by oxygen–glucose deprivation/reoxygenation (OGD/R). When there is increased oxidative stress, translocation of nuclear factor erythroid 2-related factor (Nrf2) to the nucleus leads to increased transcription of antioxidant genes. Glycogen synthase kinase-3β (GSK-3β) then phosphorylates Nrf2 and degrades it. The results showed that upregulation of PON-2 increased survival by decreasing the production of ROS, hence the oxidative stress from OGD/R. This study reinforces the idea that PON-2 has neuroprotective effects due to the enhancement of Nrf2 activity via GSK-3β phosphorylation [97].

### 3.6. PON-2 in Glioblastoma Multiforme Cell Growth

Glioblastoma multiforme (GBM) is the most malignant primary brain tumor, with the potential involvement of ROS in its tumor development. High levels of ROS can be due to a defect in the antioxidant system, which can cause impairments in gene expression and DNA repair and result in infiltrative growth of cancer. Tseng et al. found that PON-2 was highly expressed in GBM cells when compared with normal brain tissue. Moreover, when testing whether ROS decreased due to overexpression of PON-2 using flow cytometry, researchers noticed that the ROS level was significantly reduced in PON-2-overexpressed GBM cells compared with controls. In this way, GBM cells appear to use PON-2 to escape ROS-induced cell death. Valproic acid (VPA) has been used as seizure prophylaxis for patients with GBM who undergo neurosurgery. VPA decreased PON-2 expression and increased ROS production, which, in turn, activates the production of Bim, a pro-apoptotic protein. As mentioned, PON-2 may mediate an anti-apoptotic phenotype and maintain the growth of the tumor; therefore, through treatment with VPA, GBM cells face oxidative damage and cell death. Therefore, targeted reduction of PON-2 within GBM cells may potentially inhibit cancer progression [90].

### 3.7. PON-2 in Amyotrophic Lateral Sclerosis

ALS is characterized by loss of function in the cerebral cells and mitochondria, increasing in metabolic alteration, inflammation, and oxidative stress [2,98]. Oxidative stress in the central nervous system can play a central role in ALS; therefore, it is important to understand how PON-2, which has neuroprotective properties, can affect the pathophysiology of ALS. Saeed et al. analyzed the correlation between polymorphisms present in the PON gene and ALS [94]. Specifically, the C allele of the C311S PON-2 polymorphism was associated with sporadic ALS. In addition, Gagliardi et al. found that there was decreased PON-2 gene expression in spinal cord and trunk tissue of ALS patients [69], suggesting the potential involvement of PON-2 in this debilitating disease.

## 4. Overview of PON-3 and Its Neurological Associations

Paraoxonase 3 (PON-3) is the last member of the PON family of hydrolytic enzymes and is also perhaps the least studied of the three. PON-3, a calcium-dependent glycoprotein, is characterized by its anti-inflammatory, antioxidant, and anti-apoptotic properties [1]. Primarily synthesized in the liver and kidney, PON-3 is found tightly bound to HDL as it circulates the blood, in mitochondria of specific tissues, and in the endoplasmic reticulum of intestinal cells [99]. This enzyme takes on the role of hydrolyzing lactones, or cyclic esters, and eicosanoids, or signaling molecules derived from polyunsaturated fatty acids [1,99]. These characteristics are shared with PON-1, the more widely studied PON, though there are notable differences between the two family members. For example, PON-3 does not possess organophosphatase activities and its circulation is found in lower concentrations compared to PON-1 [1,98,99]. In this review we summarize some of the key studies establishing the links between PON-3 and neurological disorders, but more research is clearly needed to gain insight into these associations.

Current research suggests that PON-3 plays a role in neurodegenerative diseases that are associated with brain inflammation and oxidative lipid injury [17]. In particular, this list of diseases includes AD and ALS [17,98,99]. Other studies have also indicated that PON-3 dysfunction contributes to neurotoxicity and cerebral infarction [100,101]. Since PON-3 shares similar synthesis, expression patterns, HDL-binding in blood, and protective activities with PON-1, it is believed to exert similar effects [1,17,98,101]. Like PON-1, PON-3 dysfunction is recognized for its adverse effects in renal and cardiovascular processes [1]. After observing that PON-3 protein circulates to other regions such as the brain, researchers have studied the association between neurological diseases and PON-3 expression. Overall, a theme emerges that the neuro-associated diseases examined have some correlation with reduced PON-3 gene levels or mutated PON-3 protein. Thus, current studies of PON-3 in neurodegeneration and neurotoxicity have implications for the enzyme’s preventative and protective effects on the brain [17,98,99,100,101]. This evidence likewise suggests that, following synthesis in the liver and kidneys, PON-3 products circulate to other systems in the body and play protective roles. Ultimately, PON-3 warrants further research and thorough analysis in order to understand its localization, physiologic significance, prospects in treatment, and diagnostic potential [1,102].

### 4.1. PON-3 in Alzheimer’s Disease

HDL proteome studies suggest that loss of functionality in HDL-associated proteins, such as PON-3, may lead to AD development. While PON-3 has limited gene expression in the brain, it does circulate in various brain regions and cerebrospinal fluid [17,99]. Salazar et al. examined the PON-3 protein expression in the brain of an AD mouse model. They focused on oxidative lipid damage around amyloid-β (Aβ) plaques of specific brain cell types. Through immunohistochemistry and immunofluorescence staining of AD mouse brains, PON-3 protein levels were found to be raised in regions of high oxidative stress and in cell types, including oligodendrocytes, astrocytes, and microglia near Aβ plaques. However, in wild-type mouse models, PON-3 protein levels were scarce. Taken together, these findings have led to the conclusion that PON-3 enzymes cross the blood–brain barrier carried by HDLs, transfer to the specific brain cell types, and prevent lipid peroxidation in glial cells (oligodendrocytes, astrocytes, and microglia). Inflammation of the brain during AD pathogenesis results in ROS generation by glial cells, leading to the recruitment of PON-3 for its antioxidant function in decreasing ROS levels and reducing lipid peroxidation [17].

### 4.2. PON-3 in Amyotrophic Lateral Sclerosis

Ticozzi et al. studied the role of PON-3 DNA mutations in familial ALS (FALS) and sporadic ALS (SALS). By sequencing the coding regions of DNA, they identified the PON-3 genes in 166 FALS case samples and discovered three variants. Each novel variant was genotyped in cohorts of 1159 control, 1184 SALS, and 94 FALS DNA samples. Of the three variants, two were identified in SALS but were absent control samples, indicating a PON-3 influence in ALS development. Upon further analysis, the PON-3 variants demonstrated a high level of evolutionary conservation, emphasizing the significance of the mutated amino acids. To evaluate the impact of disease vs. control groups, the researchers found a statistically significant difference between FALS and the control groups in terms of the PON-3 variants. They found 5/260 vs. 3/1159, respectively, where *p* = 0.0069 (two-tailed Fisher’s exact test). They also investigated the statistical significance of FALS vs. SALS groups of PON-3 variants and observed that PON-3 mutations have a greater impact on FALS than SALS. The implications of these data demonstrate how loss of PON-3’s anti-oxidative properties may contribute to neurotoxic development. Because ALS pathology involves motor neuron exposure to lipid oxidation, mutations in PON-3 may disrupt the protective mechanisms against this disease development [98].

### 4.3. PON-3 in Neurotoxicity

PON-3 has also been investigated for its neuro-protective role in the brain. Almutairi et al. examined PON-3 expression levels for its potential defensive mechanisms on cisplatin-neurotoxic brains in rat models. Cisplatin, a chemotherapeutic agent, exhibits protective properties in cancer therapies, but manifests side effects of cytotoxicity in the brain. Cisplatin-induced neurotoxicity results from inflammatory cytokines, elevated ROS, and eventual lipid peroxidation. Researchers administered rutin, an antioxidant and anti-inflammatory bioflavonoid found in fruits and vegetables, to observe its preventative effect against cisplatin via the antioxidant pathway in the brain. It was found that in cisplatin-neurotoxic rats, PON-3 gene expression levels were significantly reduced compared to the control group by 4.5-fold. After administration of rutin to the cisplatin models, PON-3 levels were entirely restored to baseline levels, as in the control group, by sixfold. The data suggest that rutin provides neuroprotective effects to cisplatin-induced cytotoxicity by enhancing PON-3 expression levels. Due to its protective action of antagonizing oxidative stress and lipid oxidation, PON-3 may help to preserve nerve myelination and safeguard against neurotoxicity [100].

### 4.4. PON-3 in Cerebral Infarction

Mutations in the PON-3 gene are closely related to cerebral infarction risk (6). More commonly known as ischemic stroke, cerebral infarction results in damage to brain tissue due to a loss of oxygen [103]. One study investigated the relationship between the promoter methylator of the PON-3 gene and the risk of cerebral infarction. The promoter methylation levels of PON-3 were analyzed in 152 cerebral infarction case samples and 152 healthy control samples. Researchers observeds that PON-3 methylation was significantly lower in the cerebral infarction group compared to the control. Regression analyses revealed that heightened PON-3 methylation was associated with a protective role for cerebral infarction. These findings suggest that PON-3 hypomethylation is associated with cerebral infarction risk and that PON-3 promoter methylation could potentially be used as a prognostic marker or therapeutic target for cerebral infarction [101].

## 5. Conclusions

While the cardioprotective roles of PONs have been well established in the setting of various cardiovascular diseases, the neuroprotective functions of PONs in neurodegenerative diseases and other neurological disorders are also quite substantial across all three PON isoforms. Common themes which emerge in these settings are the ability of PONs to provide anti-oxidant and anti-inflammatory counter regulatory mechanisms in the brain which augment neuroprotection. Given the substantial morbidity and mortality associated with these neurological and neurodegenerative diseases, discovering—and developing—rational therapeutic strategies which enhance the neuroprotective effects of PONs is a logical focus of future research in this area.

## Figures and Tables

**Figure 1 ijms-24-06881-f001:**
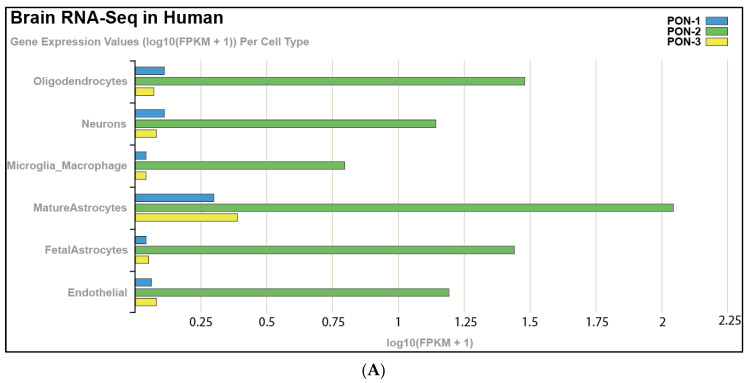
RNA-seq of cells isolated and purified from gray matter of cortex tissue, highlighting the expression of PON enzymes in different cell types in (**A**) the human brain and (**B**) the mouse brain. https://cdrl.shinyapps.io/Kaleidoscope (accessed on 19 July 2021) was used to generate figures. Cells were isolated and purified from the gray matter of cortex tissue of mouse and human brains. Cell-type-specific antibodies were used for purification: anti-CD45 to capture microglia/macrophages; anti-GalC hybridoma supernatant to harvest oligodendrocytes; anti-O4 hybridoma to harvest OPCs; anti-Thy1 (CD90) to harvest neurons; anti-HepaCAM to harvest astrocytes; and BSL-1 to harvest endothelial cells.

**Figure 2 ijms-24-06881-f002:**
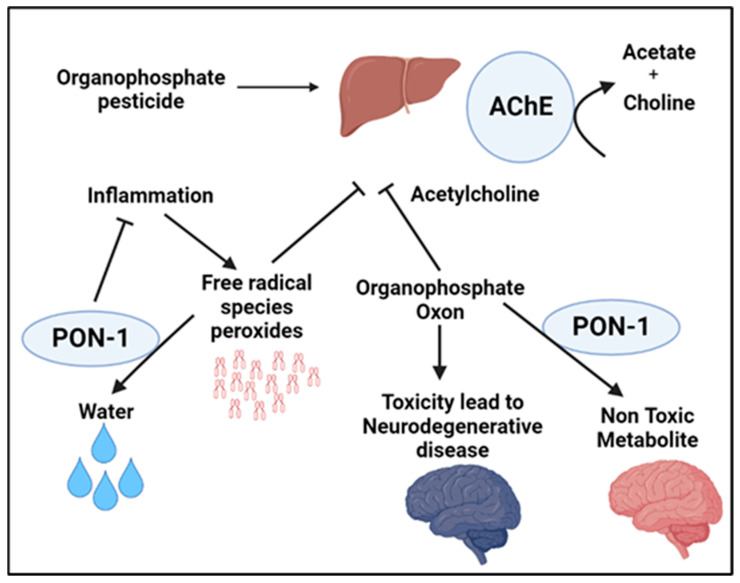
Schematic illustrating the implications of PON-1 enzymatic activity in hydrolyzing compounds associated with increased susceptibility to neurodegenerative disease, demonstrating the protective role of PON-1 in the development and progression of Parkinson’s disease.

**Figure 3 ijms-24-06881-f003:**
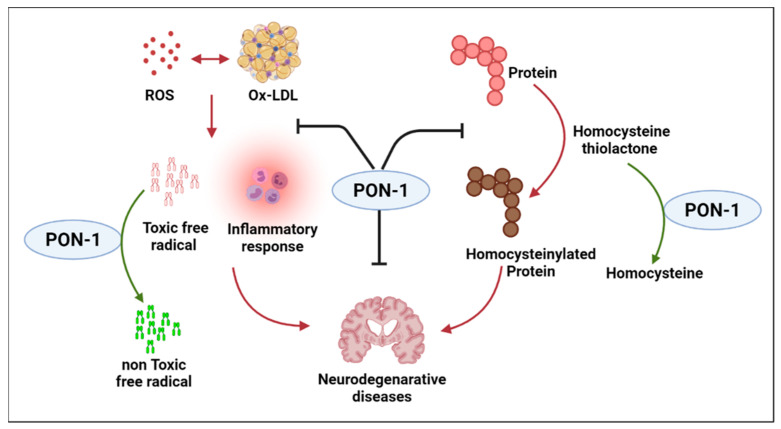
Summary of the mechanistic involvement of PON-1 against susceptibility to Alzheimer’s disease.

## Data Availability

The datasets generated or analyzed during the current study are available from the corresponding author on reasonable request.

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
