# Peer review of "Paraoxonases at the Heart of Neurological Disorders"

_ijms, 2023, doi:10.3390/ijms24086881_

Round 1
Reviewer 1 Report
The review of D.J. Kennedy and co-workers is very interesting since it gives insights on a special class of redox systems, the paraoxonases. The review is very well-written and organized, this is sure a really added-value to the literature since it synthesized up-to-date data on the implication of paraoxonases in neurodegenerative diseases. Hence, I recommend the acceptance of this manuscript. I have however some minors points,
1: Please provide a sequence alignement of the 3 paraoxonases and a scheme that describe the catalyzed reactions and structural properties with appropriate references. Indeed, the paraoxonases are not a so well-known systems, this may help to have a comprehensive overview.
2: Figures 1 to 3 are interesting but can be gathered in one or two figures, some may be put as supplementary data.
3 : Please provide a specific scheme that summarizes the molecular and cellular mechanisms behind PON's implications in AD/PD pathophysiology. This may help to summarize the most striking facts.
Author Response
Responses to the Comments
We are grateful to the reviewers and editor for the thoughtful comments and the insightful suggestions that helped us improve our manuscript considerably. As indicated in the responses below, we have taken all their valuable comments and suggestions into consideration in the revised manuscript. Please, note that Reviewers’ comments are written in black, and our responses are written in blue.
Reviewer 1
The review of D.J. Kennedy and co-workers is very interesting since it gives insights on a special class of redox systems, the paraoxonases. The review is very well-written and organized, this is sure a really added value to the literature since it synthesized up-to-date data on the implication of paraoxonases in neurodegenerative diseases. Hence, I recommend the acceptance of this manuscript. I have however some minor's points,
1: Please provide a sequence alignement of the 3 paraoxonases and a scheme that describe the catalyzed reactions and structural properties with appropriate references. Indeed, the paraoxonases are not a so well-known systems, this may help to have a comprehensive overview.
We appreciate reviewer comment and agree that it's important to provide the structural properties of the Paraoxonases enzymes, in fact our group recently published review article investigating these properties. Please refer to (A PON for All Seasons: Comparing Paraoxonase Enzyme Substrates, Activity and Action including the Role of PON3 in Health and Disease. 2022, Antioxidants, 11(3), p.590). We have cited the reference along with an added statement to highlight this important aspect. Please refer to the revised manuscript page1 highlighted section.
2: Figures 1 to 3 are interesting but can be gathered in one or two figures, some may be put as supplementary data.
We thank the reviewer for this valuable comment. We have gathered figures 1 to 3 in one figure as per reviewer recommendation. Please refer to the revised manuscript page3 figure1.
3: Please provide a specific scheme that summarizes the molecular and cellular mechanisms behind PON's implications in AD/PD pathophysiology. This may help to summarize the most striking facts.
We thank the reviewer for this insightful comment. We have modified figure 3 and 4 to highlight molecular and cellular mechanisms behind PON's implications in AD/PD pathophysiology. Please refer to the revised manuscript page 6 for Figure 2 and page 8 for figure 3.
Reviewer 2 Report
The review is potentially of great interest in the field of neurodegeneration, but the text needs to be strongly improved since in the present version it is a bit incoherent and the lecture is not easy at all. My impression is that it has been written by different authors and the different parts have been put together without being “amalgamated”. For instance, it is unnecessary to introduce Parkinson’s disease, Alzheimer’s disease, and amyotrophic lateral sclerosis every time the role of a specific PON isoform is described. Similarly, after having presented an overview of the involvement of PON1 in neurological disorders, the information does not require to be repeated in the following chapters. Again, the description of the enzymatic activity of PON enzymes in different sections is redundant and unneeded.
Moreover, while in general English is OK, some chapters need revision. I strongly suggest at least the corresponding author to read the whole text and work to make the different sections more “readable”.
The authors should also explain in more detail in which way PONs eliminate ROS as described in line #122, in Fig 5, Fig. 6, etc. As it is written and represented it appears a direct action. Is this the case? Do PONs directly remove hydrogen peroxides, superoxide anions, or hydroxyl radicals?
This is my major criticism. Other minor points are the followings:
1) the acronym HDL is first introduced at line 26 while the full name appears at line 118.
2) In figures 1, 2, and 3, acronyms must be introduced. Moreover, for the sake of clarity, in Fig 1A and 1B, as well as in Fig 3A and 3B, the x-axis must be made identical. Otherwise, it seems that PON1 RNA expression in mice is much higher than in humans, while it is the opposite, and PON3 RNA expression in mice and humans appears to be comparable, while is it expressed more in mice.
3) In line#114 does the statement “the presence of certain polymorphisms” mean that the same polymorphisms are found in different neurodegenerative disorders? If this is the case, they should be specified, and references should be added. Otherwise, the statement should be removed.
4) Q192R in line #143, but R192Q in line #167
5) Figures 4 and 5 could be merged since the information presented in Fig 4 is also partially present also in Fig. 5
6) In line #234 “physiological changes”? Maybe “pathological changes” is more appropriate!
7) in line #236 the authors state “the central role of AB in the development of AD has been largely discredited”. First, they should add some references, but then in the following sentences, the role of AB has been largely mentioned… So I do not understand exactly what the authors propose!
8) Line #351 what does the statement “ALS is heritable, though recent studies have shown that various genetic defects may contribute to familial ALS” mean?
9) In line #355 the “greatest known” should be replaced by “best known”
10) if the review focuses on the involvement of PONs in neurodegenerative disorders the chapters unrelated to neurodegenerative conditions, such as brain cancers, could be removed.
Author Response
Responses to the Comments
We are grateful to the reviewers and editor for the thoughtful comments and the insightful suggestions that helped us improve our manuscript considerably. As indicated in the responses below, we have taken all their valuable comments and suggestions into consideration in the revised manuscript. Please, note that Reviewers’ comments are written in black, and our responses are written in blue.
Reviewer 2
The review is potentially of great interest in the field of neurodegeneration, but the text needs to be strongly improved since in the present version it is a bit incoherent and the lecture is not easy at all. My impression is that it has been written by different authors and the different parts have been put together without being “amalgamated”. For instance, it is unnecessary to introduce Parkinson’s disease, Alzheimer’s disease, and amyotrophic lateral sclerosis every time the role of a specific PON isoform is described. Similarly, after having presented an overview of the involvement of PON1 in neurological disorders, the information does not require to be repeated in the following chapters. Again, the description of the enzymatic activity of PON enzymes in different sections is redundant and unneeded.
Moreover, while in general English is OK, some chapters need revision. I strongly suggest at least the corresponding author to read the whole text and work to make the different sections more “readable”.
We thank the reviewer for this very important comment and in response, we have thoroughly revised the manuscript to address reviewer's critical concern. Please refer to the revised the manuscript.
The authors should also explain in more detail in which way PONs eliminate ROS as described in line #122, in Fig 5, Fig. 6, etc. As it is written and represented it appears a direct action. Is this the case? Do PONs directly remove hydrogen peroxides, superoxide anions, or hydroxyl radicals? We appreciate the reviewer's insightful comment and agree that it's important to provide detailed explanation on PONs role in ROS elimination. In fact, our group recently published an article investigating this property. Please refer to (Khalaf, F.K., Mohammed, C.J., 2022. Paraoxonase-1 regulation of renal inflammation and fibrosis in chronic kidney disease. Antioxidants, 11(5), p.900.).
(Aviram, M., Rosenblat, Paraoxonase inhibits high-density lipoprotein oxidation and preserves its functions. A possible peroxidative role for paraoxonase. The Journal of clinical investigation, 101(8), pp.1581-1590). Our study as well as many other studies have reported direct role for PON in inflammation and oxidative stress settings. We have cited references along with an added statement to highlight this important role of PON. Please refer to the revised the manuscript page 3 highlighted section.
This is my major criticism. Other minor points are the followings:
1)The acronym HDL is first introduced at line 26 while the full name appears at line 118.
We thank the reviewer for this valuable comment. We have fixed that, please refer to the revised manuscript page1.
2) In figures 1, 2, and 3, acronyms must be introduced. Moreover, for the sake of clarity, in Fig 1A and 1B, as well as in Fig 3A and 3B, the x-axis must be made identical. Otherwise, it seems that PON1 RNA expression in mice is much higher than in humans, while it is the opposite, and PON3 RNA expression in mice and humans appears to be comparable, while is it expressed more in mice.
We thank the reviewer for this insightful comment. We have unified the x-axis as well as gathered the data in one figure for each human and mice PON enzymes expression to make it easier to compare and for the sake of clarity as per reviewer recommendation. Please refer to the revised the manuscript page 3 Figure 1A and B.
3) In line#114 does the statement “the presence of certain polymorphisms” mean that the same polymorphisms are found in different neurodegenerative disorders? If this is the case, they should be specified, and references should be added. Otherwise, the statement should be removed.
We thank the reviewer for this valuable comment. In response, we have removed the statement as per the reviewer's comment. Please refer to the revised manuscript page2.
4) Q192R in line #143, but R192Q in line #167
We thank the reviewer for this insightful comment, it should read R192 in line #167. We have fixed this accordingly.
5) Figures 4 and 5 could be merged since the information presented in Fig 4 is also partially present also in Fig. 5
We appreciate the reviewer's comment. We have merged figures 4 and 5 in one figure as per reviewer recommendation. Please refer to the revised manuscript page 6 Figure 2.
6) In line #234 “physiological changes”? Maybe “pathological changes” is more appropriate!
We thank the reviewer for this insightful consideration. We have edited the statement, now it reads pathological changes instead of physiological changes. Please refer to the revised manuscript page 6 highlighted statement.
7) in line #236 the authors state “the central role of AB in the development of AD has been largely discredited”. First, they should add some references, but then in the following sentences, the role of AB has been largely mentioned… So I do not understand exactly what the authors propose!
We appreciate the reviewer's comment, we have removed the statement to avoid confusion Please refer to the revised manuscript.
8) Line #351 what does the statement “ALS is heritable, though recent studies have shown that various genetic defects may contribute to familial ALS” mean?
We thank the reviewer for this comment we have edited the sentence for clarity. Please refer to the revised manuscript page 9 highlighted statement.
9) In line #355 the “greatest known” should be replaced by “best known”
We appreciate the reviewer's comment. We have replaced it as per the reviewer recommendation. Please refer to the revised manuscript page 9 highlighted statement.
10) if the review focuses on the involvement of PONs in neurodegenerative disorders the chapters unrelated to neurodegenerative conditions, such as brain cancers, could be removed.
We thank the reviewer for this insightful consideration. While the main goal of our review is to highlight the role of PON in neurological diseases in general, PON seems to have a key role in neurodegenerative disorders as compared to other conditions since it appears that the review focuses on the involvement of PONs in neurodegenerative disorders only. Hence, to address this valuable comment, we added a note to clearly explain the main goal of this review. Please refer to the revised manuscript Abstract page 1 highlighted statement.
Round 2
Reviewer 2 Report
The authors have considered most, but not all, of my criticism. There are still a few corrections to be made before acceptance.
1) In the first revision step I wrote " In line#114 does the statement “the presence of certain polymorphisms” mean that the same polymorphisms are found in different neurodegenerative disorders? If this is the case, they should be specified, and references should be added. Otherwise, the statement should be removed."
The authors replied " We thank the reviewer for this valuable comment. In response, we have removed the statement as per the reviewer's comment. Please refer to the revised manuscript page2.", but in the revised version the statement has not been removed (now line #101).
2) In the first revision step I indicated a typo "Q192R in line #143, but R192Q in line #167".
The authors then replied "We thank the reviewer for this insightful comment, it should read R192 in line #167. We have fixed this accordingly.", but the typo (now line #153) has not been corrected.
3) In the first revision step I wrote "in line #236 the authors state “the central role of AB in the development of AD has been largely discredited”. First, they should add some references, but then in the following sentences, the role of AB has been largely mentioned… So I do not understand exactly what the authors propose!"
The authors replied " We appreciate the reviewer's comment, we have removed the statement to avoid confusion Please refer to the revised manuscript.", but in the revised version the statement has not been removed (now line #221).
4) In addition to the previous observation, in line #507-508 of the revised version, the authors state "The development of PD is characterized by increased levels of dopamine and loss of function mutations in the DJ-1 (PARK7) gene, which can account for about one percent of PD [89]."
As it is written, the sentence is wrong! It is worth mentioning that DJ-1 mutations account for 1% of familial PD, which means that such mutations are very rare!
5) in the whole manuscript there are many typos, but I suppose that they will be corrected by the editorial staff...
Author Response
Responses to the Comments
We are grateful to the reviewers and editor for the thoughtful comments and the insightful suggestions that helped us improve our manuscript considerably. As indicated in the responses below, we have taken all their valuable comments and suggestions into consideration in the revised manuscript. Please, note that Reviewers’ comments are written in black, and our responses for the second round are written in green.
The authors have considered most, but not all, of my criticism. There are still a few corrections to be made before acceptance.
1) In the first revision step I wrote " In line#114 does the statement “the presence of certain polymorphisms” mean that the same polymorphisms are found in different neurodegenerative disorders? If this is the case, they should be specified, and references should be added. Otherwise, the statement should be removed."
The authors replied " We thank the reviewer for this valuable comment. In response, we have removed the statement as per the reviewer's comment. Please refer to the revised manuscript page2.", but in the revised version the statement has not been removed (now line #101).
We appreciate reviewer comment, we have removed the statement, please refer to the revised manuscript page 3.
2) In the first revision step I indicated a typo "Q192R in line #143, but R192Q in line #167".
The authors then replied "We thank the reviewer for this insightful comment, it should read R192 in line #167. We have fixed this accordingly.", but the typo (now line #153) has not been corrected.
We appreciate reviewer comment, we have corrected the typo now it reads R192, please refer to the revised manuscript page 4, highlighted part.
3) In the first revision step I wrote "in line #236 the authors state “the central role of AB in the development of AD has been largely discredited”. First, they should add some references, but then in the following sentences, the role of AB has been largely mentioned… So, I do not understand exactly what the authors propose!"
The authors replied " We appreciate the reviewer's comment, we have removed the statement to avoid confusion Please refer to the revised manuscript.", but in the revised version the statement has not been removed (now line #221).
We appreciate reviewer comment, we have removed the statement, please refer to the revised manuscript page 6.
4) In addition to the previous observation, in line #507-508 of the revised version, the authors state "The development of PD is characterized by increased levels of dopamine and loss of function mutations in the DJ-1 (PARK7) gene, which can account for about one percent of PD [89]."
As it is written, the sentence is wrong! It is worth mentioning that DJ-1 mutations account for 1% of familial PD, which means that such mutations are very rare!
We appreciate reviewer comment, we have revised the statement as per reviewer recommendation, please refer to the revised manuscript page 12, highlighted statement.